# OpenReview forum: "Fast Implicit Constrained Optimization of Non-decomposable Objectives for Deep Networks"
_NeurIPS.cc/2022/Workshop/HITY — HITY Workshop NeurIPS 2022_

### Official Review · Reviewer_DVaV · 2022-10-12

**Rating:** 1
**Confidence:** 4

**Review:**

This paper proposes to an efficient method for optimizing non-decomposable metrics (those which depend on multiple training points) by optimizing the NN feature extractor (the first $L-1$ layers) and the last layer separately, and assuming that the non-decomposable metric is a function of only the last layer.

I think the method is interesting and can be useful as an alternative to prior (more expensive) methods.

---

### Official Review · Reviewer_t1mk · 2022-10-13

**Rating:** 1
**Confidence:** 3

**Review:**

**Summary:** The authors propose a new training strategy for a family of
constrained optimization problems with non-decomposable objectives (e.g. false
positive rate). The main idea is to make the prediction thresholds $\lambda$ a
function of the last layer parameters only (instead of all network parameters as
in Implicit Constrained Optimization) which accelerates gradient-based
optimization.

**Strengths, Weaknesses & Questions:**
- The paper is well structured and nicely written with precise mathematical
derivations. The introduction is well-written and provides a good overview of
the context and contribution of the work. The problem, methods and experiments
are clearly described.
- Line 34-43: The *motivation* for making the prediction thresholds a function
only of the last layer's parameters is not clear to me. If I understand
correctly, this idea is motivated by works that show that the bottom layers can
be trained on noisy/imbalanced data whereas the last layer requires a
clean/balanced validation set. However, I don't see how these findings are
connected to your idea - to me, it seems that there is an argumentative gap
here.
- Table 1: I find it somewhat critical to highlight the best performance among
*your* methods in bold and not the best among all competitors. While you are
transparent about this in the caption, it might still be misleading at first
glance, suggesting that your methods always outperform the three baselines CE,
TFCO and ICO (which is not the case).
- Line 140-145: This is my main point of criticism. If I understand correctly,
the main selling point for your approach is the reduced computational costs
compared to ICO. And, while this seems plausible, you do not provide
quantitative evidence for this. So, I'm wondering *how much* faster your
approach is. My suggestion would be to report the performances in Table 1
together with the run time. You could also turn this into a 2D plot (with
runtime on one axis and performance on the other).

**Minor:**
- Line 82: Missing reference
- Equation (4), line 104: The notation is a bit confusing to me. For example,
why does $\lambda$ appear left of $\vert$ in the definition of $\tilde{l}$ since
it is not part of the minimization (over $\theta_L$) - is $\lambda$ not regarded
as fixed here? And why does $\mathbb{R}^m$ appear in the domain of $\tilde{r}$
in line 104 (the function only depends on $\theta_{-L} \in
\mathbb{R}^{p_{-L}}$)?

---

### Official Review · Reviewer_S6gH · 2022-10-17
**A modified and fast version of implicit constrained optimization.**

**Rating:** 1
**Confidence:** 3

**Review:**

The paper proposes a novel method that modifies implicit constrained optimization (ICO) in a way that the constraint influences the top layer of the network only. The reasoning the authors provide is that the top layer is associated to be responsible for the problem at hand while the bottom layers of a network provide general features. This is definitely an interesting take and simplification which merits further attention.

---

### Decision · Program_Chairs · 2022-10-20

Accept